# Emotion dynamic patterns between intimate relationship partners predict their separation two years later: A machine learning approach

**Peter Hilpert**[1]*, **Matthew R. Vowels**[1,2], **Merijn Mestdagh**[3], **Laura Sels**[4]

**1** Department of Psychology, University of Lausanne, Lausanne, Switzerland, **2** Department of Psychology, University of Surrey, Guildford, United Kingdom, **3** Department of Psychology, KU Leuven, Leuven, Belgium, **4** Department of Psychology, Ghent University, Ghent, Belgium

☯ These authors contributed equally to this work.

* peter.hilpert@unil.ch

**Data Availability Statement:** Data can be downloaded from OSF: https://osf.io/kw7nv/ DOI 10.17605/OSF.IO/KW7NV.

## Abstract

Contemporary emotion theories predict that how partners' emotions are coupled together across an interaction can inform on how well the relationship functions. However, few studies have compared how individual (i.e., mean, variability) and dyadic aspects of emotions (i.e., coupling) during interactions predict future relationship separation. In this exploratory study, we utilized machine learning methods to evaluate whether emotions during a positive and a negative interaction from 101 couples ($N = 202$ participants) predict relationship stability two years later (17 breakups). Although the negative interaction was not predictive, the positive was: Intra-individual variability of emotions as well as the coupling between partners' emotions predicted relationship separation. The present findings demonstrate that utilizing machine learning methods enables us to improve our theoretical understanding of complex patterns.

## Introduction

The degree to which partners are happy with their relationship in the long-term depends on what emotions they experience with each other during day-to-day interactions [1]. For example, experiencing many negative and few positive emotions during interactions with the partner is associated with relationship dissatisfaction [1,2]. On top of what partners feel individually during an interaction, multiple contemporary emotion theories assume that how partners' emotions are linked together across an interaction (i.e., interpersonal emotion dynamics) can also inform how well the relationship functions [3,4].

Even though previous studies have investigated whether the dynamic emotional experiences of two partners during an interaction are related to, and can predict how satisfied these partners are with the relationship, few studies have systematically compared how these intra-individual emotions and inter-individual emotion dynamics can effectively predict a future separation. As there is no one-to-one relationship between relationship satisfaction and relationship stability, it is important to extend existing research by examining emotion dynamics that determine relationship separation. Indeed, the aim of this study is primarily to determine

**Funding:** EC, PK: GOA/15/003; OT/11/031, Research Fund of the University of Leuven, https:// www.kuleuven.be/english/research/support/if EC, PK: IAP/P7/06, Interuniversity Attraction Poles programme, http://www.belspo.be/belspo/iap/ index_en.stm EC, PK, FT: G.0582.14, Fund for Scientific Research-Flanders, https://www.fwo.be/ en/ PH: P2ZHP1_151628, Swiss National Science Foundation, https://www.snf.ch PH: P300P1_164582, Swiss National Science Foundation, https://www.snf.ch PH: P3P3P1_174466, Swiss National Science Foundation, https://www.snf.ch The funders had no role in study design, data collection and analysis, decision to publish, or preparation of the manuscript.

**Competing interests:** The authors have declared that no competing interests exist.

whether in a 10-minute interaction, the emotional dynamics between the partners contain enough information to predict a later separation, by analyzing them using a machine learning approach.

## Emotions in dyadic interactions

It is widely accepted that the quality and stability of intimate relationships are largely based on the way partners interact together [5,6]. Although relationship interactions can focus on different aspects such as conflicts, positivity, or solving problems, the underlying mechanisms are always the same—relationship partners mutually influence each other through a behavioral exchange that unfolds over time [7]. However, it is not only the behavioral exchange that matters but also what partners experience emotionally during interactions. These behavioral and emotional processes between partners are indicative of how a couple functions and, therefore, predict couples' satisfaction and stability [6,8].

Emotional experiences in relationship interactions can be investigated on two different levels: the in*tra*-individual and the in*ter*-individual level. The *intra-individual* level focuses on how emotions occur within a person. Emotions are generally stimulated by relevant internal and external events [9,10]. In couple interactions, emotions are mainly based on (i) a person's own motives, needs, goals, and fears [10] and (ii) the perception and evaluation of the partner's behaviors [10,11], which in turn leads to a corresponding emotion based on evaluation of the partner's behaviors [9,10]. Therefore, emotions can contain information about how functional the interaction process unfolds, which in turn might be predictive of the couple's relationship stability.

To assess such in*tra*-individual emotions unfolding during interactions, researchers often use a video-mediated recall method [12], where the interaction is videotaped and partners watch their interaction afterward and rate their own emotional experience continuously using a wheel or slider [13]. In many of these studies, the continuous emotion reports are then aggregated, for instance into average self-reported emotions [14,15] or used to calculate a ratio between the frequencies of positive and negative emotions [15]. Findings from such studies show that these intra-individual emotions indeed matter for the relationship. For instance, individuals in unhappy relationships report higher mean levels of negative affect and lower mean levels of positive affect during interactions compared to happy couples [15–17]. Furthermore, studies on affect ratio found that partners are more likely to separate when the ratio between their positive and negative affect is less than 5 to 1 [2,5].

Other studies focused on intra-individual emotion *variability* (i.e., the average deviation from a person's mean), using experience sampling methods and daily diary methods [14] to assess how emotions vary over days but thus far the results are inconclusive [14,18]. Adolescents report faster and more extreme mood swings in comparision with adults [18] and intra-individual emotion variability is enough stable to be defined as a psychological trait [19]. However, emotion variability has hardly been investigated in the context of emotion dynamics in couples [20].

In addition to the intra-individual level, Gottman also postulated an *inter-individual emotion level* in the late 1980s [21] which has gained a lot of attention in recent years [3,4], [22]. Multiple contemporary emotion theories [3,7] suggest that the emotions that partners experience are temporally interdependent during an interaction, i.e., that the change of one person's emotion influences the other person's emotion [3]. This has also been defined as *coupling* or *coupled regulation*–how the two individually self-regulated emotion systems influence each other [23]. For instance, during a conflict, a person's increase in negative emotions might result in an expression of anger, which then influences the partner's emotions. These continuous bidirectional influences can result in feedback loops where partners mutually influence

each other [22]. As a consequence, it is expected that the interpersonal emotion dynamics between partners also contain information about the functionality of their interactions and, therefore, contain information about couples' satisfaction and stability [4].

In order to capture such inter-individual emotion dynamics based on real-time interaction data, different statistical methods have been utilized over the last decades. For example, Gottman and colleagues have used sequential analyses [17,24]. After classifying 10-second sequences as positive, negative, or neutral, the researchers predicted their partners' next sequence based on their prior sequence. They found that unhappy couples exhibit greater reciprocity of negative affect [17,24] and higher affect reciprocity compared to happy couples [12]. Because sequential analysis can only predict the subsequent sequences, Gottman and Levenson [12] utilized spectral time series analysis. By decomposing the time series, this method allows one to consider all lags simultaneously.

Thus far, findings on *inter*-individual emotion dynamics and its associations with relationship satisfaction in couples during interactions are inconclusive, sometimes revealing negative associations between the extent of observed emotional linkage for negative emotions and relationship satisfaction [15,17,22], and sometimes finding no association [25,26]. In addition, only two studies examined how interpersonal emotion dynamics predict subsequent relationship satisfaction. In those studies, it was found that relationship satisfaction declined over 3 years when wives reciprocated their husband's negative emotions [12] and that the extend to which husbands downregulate their partner's negative emotional experience predicts wives' higher future relationship satisfaction [27]. With regards to relationship stability and divorce, there are several studies examing how observed and coded behavior predicts divorce [1,28,29] but we are not aware of any study focusing on the prediction of relationship stability using self-reported moment-to-moment emotional experiences. As there is no one-to-one association between objective behavior and subjective emotional experiences, it is an open question whether this self-reported information can be used to predict couples' stability.

Research shows that although relationship satisfaction and relationship dissolutions are obviously related to each other, there is no one-to-one association between relationship satisfaction and an objective measure like divorce or relationship stability. It is crucial to understand what contributes to relationship instability because of its far-reaching individual consequences such breakup and divorce but also economically and for involved children [30]. Therefore, more research is needed to examine if inter-individual emotion dynamics can be used to predict later breakups.

Further, the methods that are most often used to examine interpersonal emotion dynamics have specific limiting requirements. Traditional statistical models require the response time between partners to be specified and fixed (e.g. 1 second, fixed sliding window). However, the influence of the emotion of one person on the emotion of the partner can vary according to the dynamics of the particular situation. For example, it may depend on how fast an experienced emotion is expressed, how fast the partner gets affected (i.e., self-regulation), and how temporally accurate the emotions are captured by an emotion wheel. The consequence of taking models that assume a specific structure and time interval concerning the reaction processes between two can lead to biased or invalid conclusions. Therefore, it is necessary to identify statistical methods which might better account for the complexity of real-time processes and simultaneously examine all lags.

## Current study

In order to investigate whether int*ra*-individual (i.e., mean, variability) and/or int*er*-individual emotion dynamics (i.e., coupling) can predict relationship stability, we utilized a dyadic

interaction paradigm, in which couples came into the laboratory to participate in a negative interaction (i.e., conflict) and a positive interaction and subsequently rated their emotions. The couples were then surveyed after two years regarding their relationship status. Thus far, evidence shows that emotions experienced in a positive and a conflict interactions are equally predictive of relationship satisfaction and stability [31].

In line with current findings on relationship satisfaction [12,27], we hypothesize exploratorily that the emotions experienced by partners during an interaction (i.e., negative and positive interaction) can be used to predict in general the couple's stability over the next two years (H1). In order to go beyond just the predictive aspect of those analyses, we also exploratorily compare whether the following aspects of intra and inter-individual emotions matter: mean, variability, and coupling. This will enable us to gain at least some insight into which aspects are important to make the prediction possible. First, based on existing research, we assume that in*tra*-individual *mean* emotion level across the interaction matters [17]: participants who are happier on average during an interaction with the partner, might be less likely to break up (H2a). Second, we hypothesize that the in*tra*-individual *variability* of the emotions during the interaction matters [20]: it might be that less emotional stability—more variability—predicts lower levels of relationship satisfaction and more breakup (H2b). Although this has not been directly tested for couples, we infer it based on findings in the field of individual emotion dynamics and well-being [14]. Finally, we predict that how the int*er*-individual emotions are *coupled* between partners, in terms of the shared rates of emotion fluctuation, predict relationship stability (H2c), as previous research indicates that some emotional linkage is related to later relationship satisfaction [12,27].

In order to overcome the methodological limitations, a stepwise approach is used to extract first features using spectral analysis [12,32] which then are analyzed with a machine learning approach known as a Random Forest (RF). The spectral part of the analysis enables us to capture the periodic fluctuations in emotion, as well as the fluctuations which are shared between partners (i.e., whether or not the two partners are 'coupled' to the extent that they have emotions which fluctuate at the same rate as each other). The RF helps us to handle high-dimensional data, and provides a data-driven, exploratory method for identifying associations between a set of predictors (the spectral information about the rates of emotion fluctuation), and the outcome (whether or not couples separated). Indeed, as we do not know *a priori* at which rates of fluctuation couples' emotions tend to vary, our choice of analytical techniques allow for multiple, possibly simultaneous, rates of fluctuation, which together represent the emotion dynamics of a couple. The spectral part of the analysis provides a set of results indicating at which rates the emotion of a partner, or the partners together, tend to fluctuate. It is for these complex, high-dimensional spectral results that the random forest is required.

## Method

### Participants

Participants lived in Belgium and were recruited through social media and flyers. In total, 101 heterosexual couples ($N$ = 202 individuals) participated in a larger study on emotions in intimate relationships (data for this study can be downloaded via OSF: https://osf.io/kw7nv/). Participants were on average 26 years old ($SD$ = 5 years, range = 18 to 53 years), had been in the relationship on average for 4.5 years ($SD$ = 2.80, ranging from 7 months to 21 years), and 14.1% were married. Most participants were Belgian (92.6%), some were Dutch (4.5%), German (1.5%), and one participant came each from Armenian, Chinese, and Ukraine. Most participants had a degree from a University (50.5%) followed high school (27.7%), higher education (21.3%), and a primary school degree (0.5%). The study was approved by the ethics

committee of the Faculty of Psychology and Educational Sciences, KU Leuven, Belgium (G-2016 02 466).Participants provided written consent to participate in this study. Couples received a compensation of 100 euros after completing the whole study.

## Procedure

The whole study consisted of different parts, but we will restrict ourselves to what is relevant here. Details about the instructions can be found in Sels et al., [26]. Couples filled out questionnaires online before coming to the laboratory. In the laboratory, they were asked to have an interaction about a negative topic (i.e. each other's most annoying characteristics) and about a positive topic (i.e. each other's most valuable characteristics), each 10 minutes long.

## Measures

**Demographic variables.** Participants reported their age, education, relationship duration, relationship status, and duration of their relationship.

**Affective experience during the interactions.** After each interaction, participants watched their video-recorded interaction on separate computers. Using a joystick, they rated their emotional experience continuously, on a second-to-second basis, while watching the interaction (left: very negative, coded as -1; right: very positive, coded as +1).

**Relationship stability.** After one and two years, participants were asked to report if they were still together with the partner from the study. After two years, 17 couples reported to have broken up. One couple was excluded as they broke up after one year but got together in year two.

## Statistical analysis

The main goal of the current study is to investigate whether intra and inter-individual aspects of emotions contain information about relationship stability within the next two years. In comparison to research from the 1980s based on couple interactions, newer statistical methods such as Vector Autoregressive Models (VAR), Dynamic Structural Equation Modelling (DSEM), and Dynamical Systems Modelling (DSM) have been used to examine how changes in one person's emotion influence the changes in the other person's emotions. These models have the advantage of not needing to sequence and categorize emotions [22,26,33,34]. In addition, VAR and DSEM models can handle linear interpersonal emotion dynamics whereas the DSM method can also handle nonlinear dynamics.

However, VAR, DSEM, and DSM require both a fixed response time between partners (e.g., 1 second) as well as the structure describing the system to be prespecified and fixed, and can be sensitive to endogeneity in the predictors [35–38]. With these methods, it is also not possible to examine data with unknown non-linear relationships or high dimensionality, owing to issues with cancellation effects and multicollinearity, respectively [36,39]. Whilst traditional methods can be adapted to handle non-linear data through the use of functions of the predictors (e.g., $x^2$), they cannot adapt to *a priori* unknown non-linear relationships. The presence of such non-linear relationships results in highly biased models. Similarly, in the presence of many input features, multicollinearity can make it impossible to estimate model coefficients.

First of all, the problem with a fixed response time must be solved. It can be assumed that the influence of the emotion of one person on the emotion of the partner varies depending on the expressivity of the situation and the expressivity of the partner. This means that the response time can vary and should ideally not be assumed *a priori*. Previous work has dealt with this issue through the use of cross-spectral analysis [12], which is a type of spectral

analysis that accounts for all possible lags simultaneously. Thus, we implement this approach because it solves this particular problem. Spectral analysis yields frequency-domain transformations of time series data, where regular cycles in the data are represented at different levels of intensity. For example, if a partner's emotion oscillates at a rate of once per minute, the corresponding spectral representation would reflect this with a peak at the associated frequency. One further advantage that spectral approaches have over typical times series approaches is that no temporal aggregation is required. In other words, the spectral representation is a true transformation in the sense that no information is lost in deriving it from the raw time-series data.

Following Vowels et al. [32,40], all data were analyzed at the level of the dyad. This was achieved by first deriving spectral features for all individuals. Then, from the spectral features for each partner in a couple, a dyadic level, cross-spectral feature is derived known as the Cross-Power Spectral Density (CPSD). In the case of dyadic data, the CPSD provides an estimation of correlation between the spectral representations of each of the partner's spectra, such that two partners that fluctuate at the same rate would have a peak in the CPSD representation at the corresponding frequency. Thus, this is how we conceptualize coupling as CPSD provides information about the coherence and synchrony between partners over time. For example, if both partners fluctuate in their emotion twice per minute, then there would be peaks in both their frequency spectra at a frequency of twice per minute. This shared rate of fluctuation results in a correlation between the two spectra (i.e., the coupling), which is, in turn, represented as a peak in the CSPD.

Unfortunately, CPSDs yield high-dimensional features. In the present case, the data were collected at a rate of once per second for ten minutes, which yields 600 time points per partner and a corresponding CPSD feature which is 301 dimensions per couple [41]. As described above, however, traditional approaches such logistic regression cannot handle the full CPSD data due to issues relating to multicollinearity, and in order to solve this issue, we use a machine learning approach known as a Random Forest (RF), which is a type of data-driven decision tree [35]. As well as handling high-dimensional data, the use of RFs brings a further advantage, in that it has a flexible, data-driven functional form (i.e. the form describing the system does not need to be prespecified) and can discover highly non-linear relationships and complex interactions. Given the complexity and the possibility of non-linearity in interpersonal emotion dynamics, RFs present an effective solution. It provides us both with a tool for prediction as well as for identifying associations present in the data.

One of the possible pitfalls of such flexible models is that they have the potential to *overfit* the training sample. Overfitting is a phenomenon whereby a model exhibits good performance on the data with which it was fit, but not on unseen data from the same distribution. It therefore closely relates to generalizability, as a model that overfits the data will have poor generalizability. Random Forests are fit on bootstrapped sub-samples and thereby naturally mitigate issues with overfitting, but given their flexibility, this is not enough. Therefore, we used k-fold cross-validation [42] which is a model-fitting process involving the division of the dataset into *k* equally sized, disjoint, randomly selected splits. The model can then be iteratively fit using (*k-1*) splits and tested on the remaining split, *k* times. The advantage of *k*-fold cross-validation over a simple train-test split is that it enables all the data to be used, whereas a train-test split only enables testing on a small proportion (e.g. 25%) of the dataset thereby limiting the reliability of the results.

The final model performance is taken as the aggregate across all *k* test splits. By always evaluating the model on an unseen data split, k-fold cross-validation helps to avoid overfitting and thereby provides an estimate of performance that is more generalizable than one which is derived from a model which has been both fit and tested on the same data. It is worth noting

that Random Forests can be used to provide their own estimates for unseen data, and these are known as out-of-bag (OOB) estimates. However, there is some evidence that OOB estimates may be biased and/or different to the estimates derived using the standard k-fold cross-validation process [43,44]. In order to avoid any such issues or concerns we chose to use k-fold cross-validation.

Random Forests can be sensitive to hyperparameter settings (where hyperparameters are algorithmic settings which include attributes such as the maximum depth of the decision tree, or number of estimators). However, tuning the parameters requires an additional validation data split, which reduces the quantity of data available for training and testing. We therefore used the default hyperparameter settings in the *scikit-learn* package Random Forest [45] and thereby avoided the need for a separate validation set.

Even though the use of k-fold cross-validation helps us to avoid overfitting, which can especially be a problem for data of limited samples (in the present case, the number of people who broke up was only 17), it is nonetheless possible that the resulting performance is sensitive to the randomly chosen splits used to select the k folds. Furthermore, the Random Forest itself can converge differently each time it is trained. Thus, there exist multiple sources of performance variation which can make the resulting performance scores for each model fluctuate. In order to account for (1) potential sensitivity associated with the random splits chosen at the start of the k-fold cross-validation process, particularly with respect to the small number of people that broke up, and (2) variation in Radom Forest convergence, we repeat the entire training and testing process 50 times. This results in a distribution of model prediction/performance scores, and therefore a set of averages and standard errors for each model, which can then be compared.

**H1.** The first model (1) is a Random Forest fit to the CPSDs derived from the original time series emotion data, as well as the means and variances of each partner's time series. This model provides a means to evaluate the predictive content of dyadic, real-time emotion data. For completeness, we also compare the machine learning approach against a more traditional model. Following the same cross-validation procedure described above, we fit and evaluate a logistic regression model with a simplified data structure (i.e., mean and variance for each partner, cross-correlation for both partners) in order to provide us with a baseline level of predictive performance (results for this model can be found in the appendix).

**H2.** Even though Random Forests can be interrogated to identify which dimensions in the input feature are being used to make a prediction [35], the CPSD itself is 301 dimensional, and thus the results are not readily interpretable. To make interpretation more tractable, we remove various combinations of predictors from the data (we refer to each predictor set variant as a surrogate), and compare the associated model performance. If the predictability of the model drops, we can conclude that the information removed had some predictive value. Concretely, we create five surrogate time series before deriving the CPSDs and compare the performance of corresponding Random Forests. The five surrogates differ in their combination of the following preprocessing steps: (2a) whether they are mean-centred (averaged individual differences are removed), (2b) whether they are standardized (averaged individual differences and variances are removed), and (2c) whether individual emotions are randomly permuted in time (removing the linkage between both partners' emotions). There are five surrogates in total because we use the following combinations: mean-centred (a); standardized (b); time randomized (c); mean-centred and time randomized (a+c); and standardized and time randomized (b+c). These surrogates can be visually compared against the original model (1; in blue) as well as against each other (in green). A non-overlap of the standard error indicates a significant difference between models.

If the performance drops when we individual/person mean center the time series by subtracting the mean level of each individual's time series from their time series, compared to

model 1, then we can infer that the mean level of emotion was of predictive value (a). Accordingly, if the performance declines when we standardize the time series, by additionally dividing each individual's time series by the variance of their time series, we can infer that the variability in the emotion is of predictive value (model b). Finally, if the model performance drops when we randomly permute (i.e. sample without replacement) the order of the time series separately for each partner, thereby destroying any auto-correlation and cross-correlation between partners, then we can infer that time-based patterns reflected in the CPSD (i.e. the couple's synchrony and dynamic coupling) is predictive (c). For completeness, we also investigate different combinations of these models. This enables us to compare, for example, a mean centered model (a), with a mean centered and time randomized model (a + c). We would expect the worst performing model to be one which is standardized and time randomized (b + c).

As it is possible that emotional experience and dynamics between partners may capture redundant information, all models are compared with extended models that include demographic factors as predictors. This helps us determine whether emotions may capture redundant information.

In total, we undertake each variant of the analytical procedure, as outlined above, 50 times in order to establish a distribution over model performance which accounts for performance variability resulting from the time-randomization of the time series, the $k$-fold splitting, and the Random Forest model bootstrapping processes. Such a process provides an intuitive alternative to the use of $p$-value for evaluating and comparing different models' performance, because it provides a complete distribution over each model. Any differences between these distributions are therefore significant by definition, although care must be taken when generalizing these results to new empirical samples. We provide the mean performance over these distributions, and the associated standard errors.

We evaluate each of the Random Forest models' abilities to predict relationship stability. Random Forest classifiers are used to predict whether the couple breaks up after two years, and Matthew's Correlation Coefficient (MCC) is used as the metric of performance. The MCC [46] ranges between minus one and plus one, and provides a metric for classification performance which is robust to class imbalances and which can be interpreted in much the same way as the Pearson Correlation Coefficient. All analyses were computed with Python 3.7 [47]. Finally, to investigate whether couples who had broken up had a higher or lower magnitude of coupling than those who stayed together, we evaluated the (log) magnitude of the CPSD at different rates of fluctuation.

## Results

### Descriptive statistics

From the originally 101 couples joining the study, emotion data was successfully collected from 99 couples in the laboratory using a video recall task. One couple was excluded as they broke up during year one, but got together in year two, resulting in a final sample of 98 couples. Thus, the data set was based on 98 couples * 2 partners * 2 interactions * emotional reports in terms of valence assessed 600 times, resulting in a data set of 235,200 data points. Over the time period of two years, 17 couples broke up (17.3%). Overall, couples reported high average level of relationship satisfaction (mean$_{male}$ = 5.96, SD$_{male}$ = 0.71, mean$_{female}$ = 5.93, SD$_{female}$ = 0.73, range = 1–7). In order to provide a baseline level of predictability, we used a logistic regression model and a simplified data structure. Results show for the positive interaction an MCC of 0.059 (SE = .008), and for the negative interaction an MCC of 0.019 (SE = .008). The MCCs can later be compared with the MCCs from the Random Forest Models.

**Hypothesis 1.** Model 1 allowed us to examine whether information in the emotion data assessed during a negative and a positive interaction can be used in general to predict relationship stability over a time span of two years. The MCC results (including standard error bars) for the first model (1) are shown in Fig 1. Whilst the classifier was unable to predict relationship stability using the negative interactions, the positive interaction yielded a small but clear

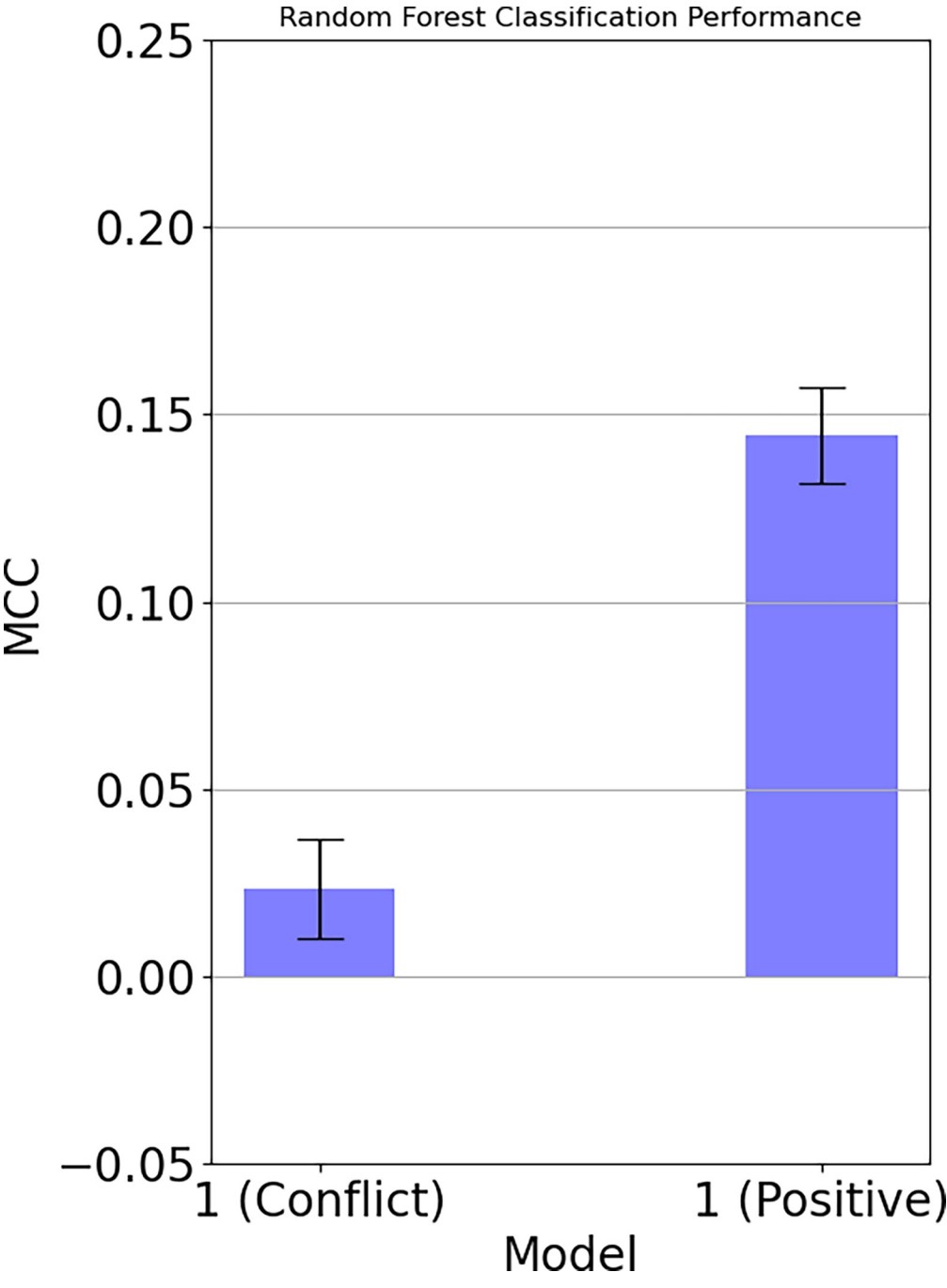

**Fig 1.** Predicting relationship stability based on negative interaction (on the left) and positive interactions (on the right). Results indicate that relationship stability could only be predicted by the emotion data from the positive interaction.

signal for predicting relationship stability. At least based on the positive interaction, the random forest was able to predict relationship stability from couples' emotions (MCC = 0.1), thereby providing an evaluation of the first hypothesis. This is in contrast to a finding showing that positive and negative interactions contexts predict relationship satisfaction [31]. However, the difference in results might also be explained by the differences in designs. Graber et al. [31] have used an average score of affect to predict relationship satisfaction within 12–15 months after the interaction whereas we examined a more nuanced process—how emotion dynamics between partners predict a breakup.

**Hypothesis 2.**  In order to evaluate the second hypothesis, we examined whether removing information of the mean (2a), variance (2b), and coupling (2c) was associated with a drop in the ability to predict relationship stability. This enabled us to contrast models 2a, 2b, and 2c with the prediction of the first breakup model (1), as well as between the surrogate data set (e.g., 1+2c; 2a+2c). The results for the negative interaction are shown in Fig 2, where it can be seen that all results indicated an absence of predictive power. This is to be expected–if

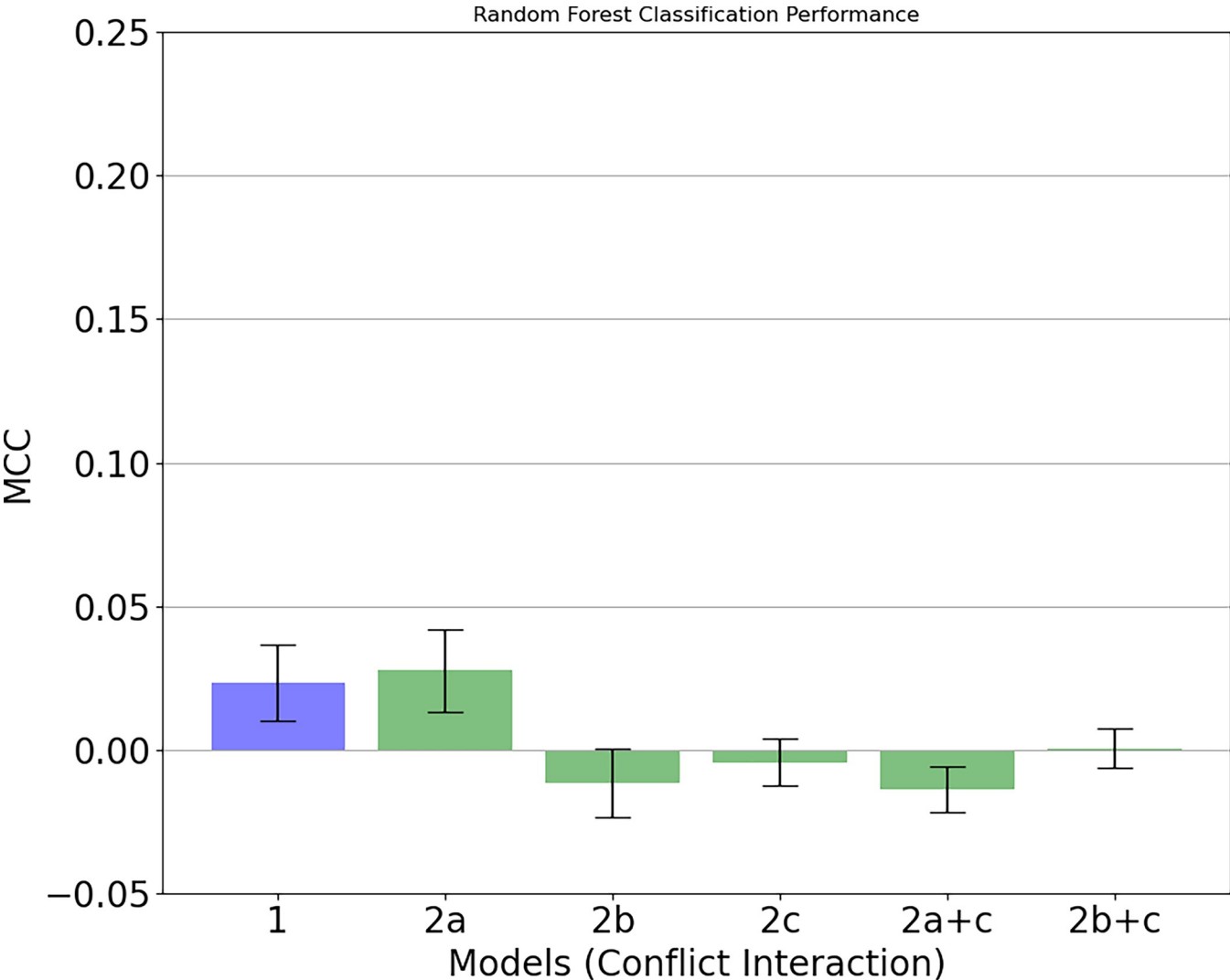

**Fig 2.** As we were unable to predict the breakup with model 1 based on the negative interaction (left bar), removing information such as mean, variance and coupling had no meaningful effect on the predictive power.

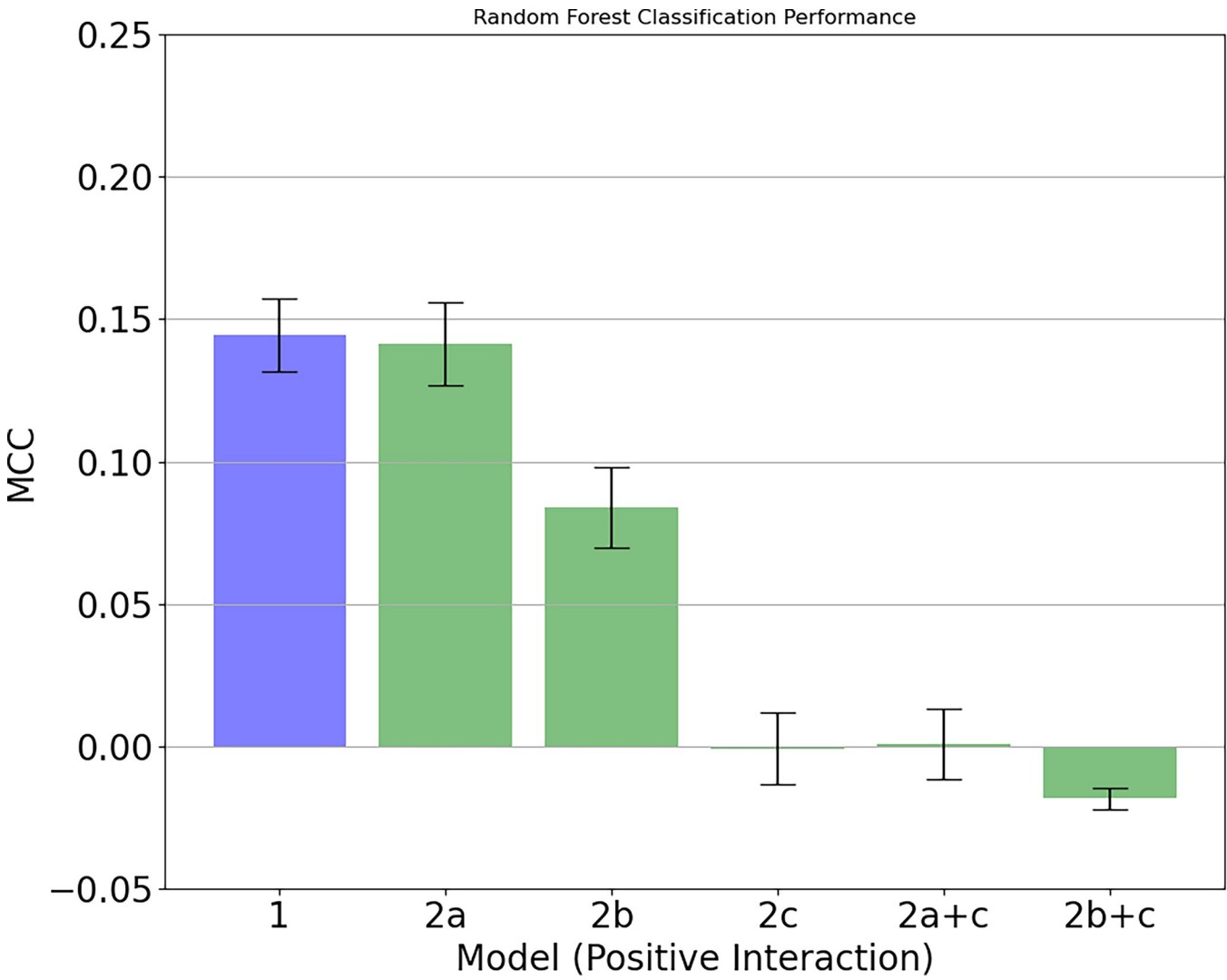

**Fig 3. The left bar indicates the prediction for relationship stability for the positive interactions (Model 1).** In comparison to this, the other bars indicate that mean centering leads to hardly any change, whereas removing the variance and the coupling (i.e., time ordering) have strong effects.

relationship stability cannot be predicted using the full data, then the removal of the mean (2a), variance (2b), and autocorrelation (2c) should not increase the performance.

As we were able to predict relationship stability from the emotions partners experienced during the positive interaction, removing information (mean, variance, time) should provide interpretable insights (Fig 3). Results show that there is a small, but non-significant drop when removing the mean information (1 vs 2a), suggesting that the mean was surprisingly not particularly important, above and beyond the variance information and the dynamic coupling between the partners.

Removing also the variance in addition to the mean (1 vs 2b) did result in a large drop in performance. This indicates that the variability of emotions across the interaction did contain important information about whether or not couples will break up. As the variance information is at the individual level, rather than the couple level, it is non-trivial to identify whether, for instance, higher or lower variance is associated with breakup. For example, it may be that partner A has high variance, and partner B has low variance. Additional point-biserial

**Table 1. Point-biserial correlations examining whether the emotional variance of someone's time series correlates with whether they break up.**

| Variance | Break-up | p-value |
|---|---|---|
| Male, positive interaction | 0.10 | .310 |
| Female, positive interaction | -0.06 | .567 |
| Male, conflict interaction | 0.18 | .086 |
| Female, conflict interaction | 0.16 | .122 |

correlations were computed between the individual emotion variability over a sequence and the breakup (Table 1). As the correlations were not significant, the direction of the relationship is not clear, indicating that variability alone is not sufficient in determining breakup. Further research is needed to specifically investigate the nature of the relationship between emotion variance and breakup.

Furthermore, we analyzed the emotion dynamics across partners (i.e., coupling; Fig 3). We compared the naturally unfolding dynamics with the surrogate data (time-randomized 1+2c) and the models where mean and variance was removed as well (2a+2c and 2b+2c). Results show the largest drop in performance. This consistent performance drop is a strong indication that the Random Forest was leveraging information about the emotion coupling between partners', captured in the CPSD features.

To investigate whether couples who had broken up had a higher or lower magnitude of coupling (i.e. a lower log-magnitude of the CPSD) than those who stayed together, we evaluated the (log-) magnitude of the CPSD at different rates of fluctuation. Fig 4 presents the histograms for the magnitude of the CPSDs at different frequency bands, for couples that broke up (grey) and couples that stayed together (blue). Across all four sub-bands, it can be seen that couples that stayed together had a higher average magnitude CPSD than those that broke up, suggesting that coupling may be a positive feature indicative of longevity in relationships.

Finally, all models were then compared to models in which we included additional information regarding demographics. However, since there were no significant differences, these results were included as supplemental material.

## Discussion

Several theories assume that both the individual emotional experience [15,17] as well as the coupling of emotions between the two partners during an interaction [14,20] provide important information about the functionality of the interaction. However, the current state of research is inconclusive as to whether the emotion dynamics that unfold during a brief interaction between partners contain any meaningful information about the current state of the relationship [25,26]. The current paper aimed to address these issues and provides some evidence that emotion dynamics between partners matter,

### Predicting relationship stability (H1)

We first hypothesized that the short-term self-reported emotional experiences of partners during interactions contain information about the functionality of the interaction and therefore enable us to predict the stability of the couple in the long-term. Regarding the interaction on a negative topic, no relationship was found. However, for the positive interaction topic, we were able to predict relationship stability from their self-reported emotional experience during the interaction. Even though the effects are small, as can be expected based on just assessing emotions for 10 minutes in the laboratory, results indicate that there is information in the short-

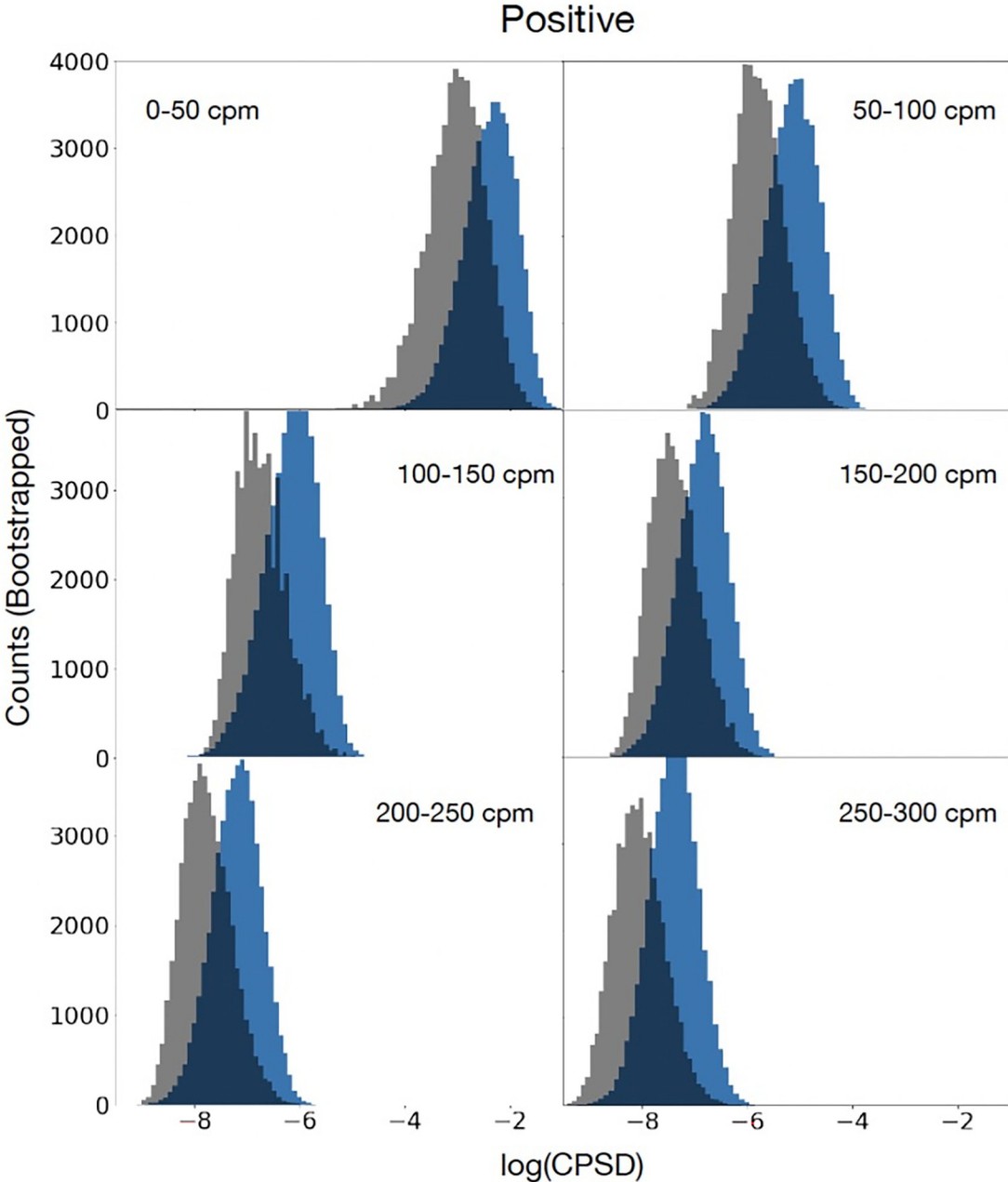

**Fig 4. Depicts bootstrapped histograms for the average log of the magnitude of the CPSD over six sub-bands for the positive interaction.** Each sub-band represents a range of fluctuation rates. For example, 0-5cpm represents fluctuation rates between zero and five cycles in a minute. We present two histograms in each subplot, one for the group of couples who broke up (grey) and one for those that stayed together (blue). Note that the couples who stayed together (consistently the histogram towards the right hand side of each subplot) have higher log-magnitude-CPSD than those that broke up (consistently towards the left hand side of each subplot). Given that CPSD measures a type of coherence in fluctuation, this suggests that couples who stayed together exhibited higher coherence in their fluctuations than those that broke up, over all sub-bands.

term emotional experience that can predict a break-up longitudinally. One explanation of this finding is that there can be both functional and dysfunctional dynamic emotional processes during an interaction. These processes seem to be so substantial that they can be used to predict a breakup two years later. In relation to the inconclusive findings [26], the evidence from this study tends to support the importance of emotion dynamics.

However, it is unclear as to why the emotional experience during the negative interaction topic had no predictive value. One could speculate that couples are reluctant to argue in front of the camera and therefore their emotional experience was less intensive as in a more natural setting at home with no one around. Indeed, couples reported to experience on average positive emotions during the conflict interactions (male = .14; female = .11; range -1/+1). And, if the interaction was not heated, we might not be able to predict whether the couple will stay together. The alternative explanation is, of course, that the emotional experience in a negative interaction context does not actually contain any information regarding the functionality of the dispute. However, this would contradict extensive past research that did find evidence for the importance of emotions in relationship functioning [1,28,29].

## Emotions: Mean, variability and coupling As mechanisms (H2)

Just because we can predict a later separation in the positive interaction context (H1), we do not know explicitly *which* aspects of the emotions enable the prediction. In this regard, the machine learning approach is a black box. Because ideally, we want to be able to make evidence-based suggestions on what couples can do to improve their relationship, we explored three aspects of emotions to give us insight into the underlying mechanisms. Specifically, we investigated the role of two *intrapersonal* emotion dynamics, this is (H2a) the *mean* level of the emotions experienced over the entire interaction, and (H2b) the *variability* of the emotions. We also investigated interpersonal emotion dynamics (H2c) *between* partners in the sense of coupling. After removing the *mean value* of the emotional experience for each person from the analysis (i.e., centering), the prediction hardly changed. This suggests that the mean emotional experience during a positive interaction context does not play a role in predicting relationship stability (H2a). This is surprising because it is plausible to assume that if someone experiences a lot of positive emotions on average during positive interaction contexts, the interaction style is functional and the relationship is experienced as satisfactory, resulting in an increased likelihood to stay in the relationship. Our finding also contrasts with other studies that examined the role of partners' average emotion levels in relationship satisfaction. For example, average emotions during interactions are associated with relationship dissatisfaction [15–17]. However, relationship stability is not determined by relationship satisfaction only [48]. Additionally, these older studies used different methodologies and statistics and focused often on negative interactions opposed to positive contexts.

As a second factor, we examined the role of *emotion variability* during the positive interaction context (H2b). Our results show that the removal of the emotional variability between people (after standardizing) led to a significant decrease in predictive power. This indicates that the amplitude of the fluctuation is actually informative and is important for predicting the separation of the couple. In contrast to the mean emotion, partners' emotional variabilities during a 10 minutes interaction was important in predicting relationship stability two years later. As this is the first study in couple research examining the effect of emotional variability, it is not yet clear what emotional stability means during such an interaction and whether this is functional. Although we can speculate based on findings on individual emotion dynamics [14] that low emotional variability is generally positive during an interaction with the partner, it might also be functional if one can emotionally react to the partner, especially during a positive interaction. Unfortunately, at the moment it is not possible for us to say whether more or less variability is important for predicting the separation because the patterns we further explored were not clear; and future studies are necessary to unravel its meaning.

Finally, we examined whether *coupling* between partners' emotions contains information that helps us to predict whether a couple will break up or not. After we randomly shuffled the emotions of each person individually according to time, the prediction of the breakup sunk to

practically zero, independent of whether we additionally removed the other two factors (mean, variance) in the model or not. This indicates that the way how partners' emotions are coupled might be an important indicator of the functionality of partners' interaction dynamics, which enabled us to predict breakup.

In order to more precisely understand the role of coupling in relationship stability, we also explored whether coupling was higher or lower in couples who broke up or stayed together. Couples who stay together exhibit higher CPSD, indicating a stronger degree of coherence. In other words, couples that stay together to more closely share fluctuations. The histograms of CPSDs at different shared fluctuation rates consistently support this expectation—couples who stay together exhibit stronger coherence across all rates of fluctuation compared to the couples who then separated two years later. This also seems conceptually plausible. In a positive interaction context in particular, one can assume that it is good if the partners are emotionally connected—if one person feels good and makes some fun and happy comments, the other partner gets emotionally affected. It is notable that we found this effect consistently across all rates of fluctuation (all possible lags), which is conceptually interesting because the emotional connection is apparently an important part of the relationship-based interaction and of couples stability.

In summary, it seems that sophisticated statistical methods such as machine learning algorithms can help uncovering the role of complex psychological interpersonal processes such as emotion dynamics in predicting long-term consequences. Continuous emotion data is difficult to assess and even more difficult to analyze with traditional statistics. Machine learning methods can overcome some of the limitations of conventional approaches and can help push our understanding of emotional processes in intimate relationships forward.

## Limitations

Despite the various strengths of this study, such as the experimental design, emotions being assessed continuously during dyadic interaction, and the use of machine learning methods, there are several limitations. Although the sample for an experimental study with couples in the laboratory is relatively large, it is relatively limited for predictions with machine learning methods. In these two years we only have 17 couples who have split up. It is therefore important to note the exploratory nature of this work, and that that larger studies will be necessary to replicate the findings. We hope that, in spite of the sample size, similar techniques to those used in this work (such as Random Forests) can be exploited in order to better understand the complex processes of emotion dynamics. Furthermore, removing the mean might not actually remove all the information about the mean (e.g. ceiling effects may result in indirect characterizations of the mean level before centering). Nevertheless, if some information about the mean would still be in the data and matter for the prediction, we would not see a drop to zero when removing the information about variance and time. It is also possible spontaneous changes in conversation topic could lead to an inflation of the measured synchronicity which does not derive from the coregulation process. As we do not have the means to distinguish whether a change in topic occurred spontaneously or as part of a coregulatory process, it was not possible to control for these kinds of event. Finally, the following demographic variables were not assessed: race, income, and socioeconomic status.

## Supporting information

**S1 File.** A) Posi4ve interac4ons versus conflict interac4ons. B) Figures for Posi4ve interac4ons. C) Figures for Conflict interac4ons.
(PDF)

## Author Contributions

**Conceptualization:** Matthew R. Vowels.

**Data curation:** Laura Sels.

**Formal analysis:** Peter Hilpert, Matthew R. Vowels, Merijn Mestdagh.

**Methodology:** Peter Hilpert, Matthew R. Vowels, Merijn Mestdagh.

**Supervision:** Peter Hilpert.

**Writing – original draft:** Peter Hilpert, Matthew R. Vowels.

**Writing – review & editing:** Peter Hilpert, Matthew R. Vowels, Merijn Mestdagh, Laura Sels.

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
