## [Decision Letter · Decision Letter 0]

6 Mar 2023

PONE-D-22-33994Emotion Dynamic Patterns Between Intimate Relationship Partners Predict Their Separation Two Years Later: A Machine Learning ApproachPLOS ONE

Dear Dr. Hilpert,

Thank you for submitting your manuscript to PLOS ONE. After careful consideration, we feel that it has merit but does not fully meet PLOS ONE’s publication criteria as it currently stands. Therefore, we invite you to submit a revised version of the manuscript that addresses the points raised during the review process.

We look forward to receiving your revised manuscript.

Kind regards,

Joydeep Bhattacharya

Academic Editor

PLOS ONE

Journal Requirements:

2. Please provide additional details regarding ethical approval in the body of your manuscript. In the Methods section, please ensure that you have specified the name of the IRB/ethics committee that approved your study

3. Peer review at PLOS ONE is not double-blinded (https://journals.plos.org/plosone/s/editorial-and-peer-review-process). For this reason, authors should include in the revised manuscript all the information removed for blind review

"EC, PK: GOA/15/003; OT/11/031, Research Fund of the University of Leuven, https://www.kuleuven.be/english/research/support/if

EC, PK: IAP/P7/06, Interuniversity Attraction Poles programme, http://www.belspo.be/belspo/iap/index_en.stm

EC, PK, FT: G.0582.14, Fund for Scientific Research-Flanders, https://www.fwo.be/en/

PH: P2ZHP1_151628, Swiss National Science Foundation,  https://www.snf.ch

PH: P300P1_164582, Swiss National Science Foundation,  https://www.snf.ch

PH: P3P3P1_174466, Swiss National Science Foundation,  https://www.snf.ch".  

Additional Editor Comments:

I had trouble finding suitable referees which explains the delay in getting to a decision. The one report I have in possession is very positive about the paper. My own quick reading also left me with a positive impression. I encourage you to revise the paper based off the comments of the referee which seem doable (esp. the part about showing circumstantial evidence about causation).

Reviewers' comments:

Reviewer's Responses to Questions

**Comments to the Author**

1. Is the manuscript technically sound, and do the data support the conclusions?

Reviewer #1: Yes

2. Has the statistical analysis been performed appropriately and rigorously? 

Reviewer #1: Yes

3. Have the authors made all data underlying the findings in their manuscript fully available?

Reviewer #1: Yes

4. Is the manuscript presented in an intelligible fashion and written in standard English?

Reviewer #1: Yes

5. Review Comments to the Author

Reviewer #1: The paper addresses a relevant question and is methodologically sound. I would recommend accepting the paper provided the following revisions are adequately addressed.

Major

The paper would benefit from showing additional benchmarks. For example, it is likely that demographic features and their interactions are predictive of relationship outcomes. Other information that was collected about participants could also be used to construct theory informed benchmarks. It would be interesting to see whether the emotion features would incrementally improve predictive performance when added to the same model (i.e. whether they capture non-redundant target variance). Being able to show that the emotion features capture non-redundant information would make the paper much stronger.

The directionality of the association between emotion variability and relationship stability remains unclear and the authors suggest this as a direction for future research. I strongly recommend addressing this question in the current paper. This could easily be done by showing descriptive relationships between properties of the distribution of input features and the target variable. For example, the authors could compute point-biserial correlations between the person-level variance scores of emotion measures and the relationship outcome. For completeness this could also be done for the mean. Alternatively, the authors could show differences between the distributions of relationship outcome classes the same way it was done for CPSD (fig 4).

Minor

MCC is a relatively uncommon measure for classification performance. For completeness the authors should add the standard metrics (accuracy, precision, recall, and AUC) to the visualizations and consider reporting AUC as the main metric throughout the paper.

The authors refer to their study as an experimental study on several occasions (e.g. page 21). This is imprecise as the study is clearly correlational and there was no randomly assigned manipulation. The authors should correct this.

Formatting / style

The following sentence is incomplete: “After classifying 10-second sequences as positive, negative, or neutral, they predicted partners’ next sequence the person’s prior sequence and found that unhappy couples experience greater reciprocity of negative affect (17,24) and high affect reciprocity 75 (12) than happy couples.”

The title on page 1 is different from the title on the cover page

6. PLOS authors have the option to publish the peer review history of their article (what does this mean?). If published, this will include your full peer review and any attached files.

Reviewer #1: No

---

## [Author Response · Author response to Decision Letter 0]

15 Jun 2023

PONE-D-22-33994

Dear editor and dear reviewer, 

Thank you very much for the suggestions. We re-run all the analysis and added the sociademographic variables, which were availeable to us (this was limited as this is a secondary data analysis). We tried to implement the comments as good as possible and hope that this has improved the manuscript. 

Review Comments to the Author

Reviewer #1: The paper addresses a relevant question and is methodologically sound. I would recommend accepting the paper provided the following revisions are adequately addressed.

COMMENT 1

Major

The paper would benefit from showing additional benchmarks. For example, it is likely that demographic features and their interactions are predictive of relationship outcomes. Other information that was collected about participants could also be used to construct theory informed benchmarks. It would be interesting to see whether the emotion features would incrementally improve predictive performance when added to the same model (i.e. whether they capture non-redundant target variance). Being able to show that the emotion features capture non-redundant information would make the paper much stronger.

ANSWER 1

Thank you for this very interesting question. In fact, creating a comparison standard is an excellent way to test whether the variability in emotions plays a role or whether emotions could potentially be a proxy for other factors. 

Therefore, we first calculated all models with the additional variables age (of each partner in the couple) and relationship living status (married, cohabiting, not cohabiting) to compare whether there was any difference, with the idea that if there were differences, we could proceed step by step to see which factors improve performance. More variables could not be included as this is a secondary data analysis and these are all the socioeconomic data we have access to. 

However, the results show that the additional variables do not lead to a change in the results. In fact, we see some reduction in predictive performance. Whilst in our experience it is more common for the addition of information to improve predictive performance, we have sometimes found that, for datasets with small effects, the relationship is not always that of a monotonic improvement. In any case, the results indicate that the effect is explained by emotional experience and dynamics between partners, and that emotions contain non-redundant information. It could have been the case that emotions were only a proxy for other factors, but the additional analyses indicate that this is not the case. 

Here are all the examples, which will be included into the supplementary material (on the right sight all are the figures including additionally the two demographic variables. 

Unfortunately, we could not include the plots here. But the plots and the comparison can be found in the supplementary material

A) Positive interactions versus conflict interactions 

Random Forest MMC vs Random Forest MMC & demographics

Random Forest AUC vs Random Forest AUC & demographics

Overall, it appears that adding demographic variables has resulted in a slight decline in prediction accuracy, rather than an improvement.

B) Figures for Positive interactions

Logistic Regression vs Logistic Regression & demographics

Random Forest vs Random Forest & demographics

Random Forest & AUC vs Random Forest & AUC & demographics

C) Figures for Conflict interactions

Logistic Regression vs Logistic Regression & demographics 

Random Forest vs Random Forest & demographics 

Random Forest & AUC vs Random Forest & AUC & demographics

It is clear that the demographic variables had no significant influence on the results. All these plots can be found in the supplemental materials. 

This reinforces our confidence in the findings. It is actually quite remarkable that emotional information from just 10 minutes of interaction can predict separation two years later. We would not argue that the observed emotions cause separation, but it may serve as a proxy for how partners interact and experience each other, which is indicative of a potential future separation, and this relationship seems worthy of future replication and exploration.

We have added this clarification in the article.

Statistical Analysis

“As it is possible that emotional experience and dynamics between partners may capture redundant information, all models are compared with extended models that include demographic factors as predictors. This helps us determine whether emotions may capture redundant information.”

Results

“Finally, all models were then compared to models in which we included additional information regarding demographics. However, since there were no significant differences, these results have been included as supplemental material.”

COMMENT 2

The directionality of the association between emotion variability and relationship stability remains unclear and the authors suggest this as a direction for future research. I strongly recommend addressing this question in the current paper. This could easily be done by showing descriptive relationships between properties of the distribution of input features and the target variable. For example, the authors could compute point-biserial correlations between the person-level variance scores of emotion measures and the relationship outcome. For completeness this could also be done for the mean. Alternatively, the authors could show differences between the distributions of relationship outcome classes the same way it was done for CPSD (fig 4).

ANSWER 2

Thank you for this idea. We have already attempted to go beyond pure prediction and understand conceptually whether more or less variance predicts separation. This would be interesting to know because exploratively we could draw the conclusion whether too much emotional fluctuation or too little fluctuation might have something to do with later separation, even though this would only be on an exploratory level.

As suggested, we have examined the biserial correlations to see if there are any relationships between properties of the distribution of input features and the target variable. However, the results of the biserial correlations indicate no significant relationship between the variance of someone's time series and the breakup. All p-values are greater than 0.05. The result supports our previous statement that the reason for the relationship is unclear, and that further research is needed.

We have supplemented the manuscript with the table of correlations and provided a more detailed description.

«This indicates that the variability of emotions across the interaction did contain important information about whether or not couples will break up. As the variance information is at the individual level, rather than the couple level, it is non-trivial to identify whether, for instance, higher or lower variance is associated with breakup. For example, it may be that partner A has high variance, and partner B has low variance. Additional point-biserial correlations were computed between the individual emotion variability over a sequence and the breakup (Table 1). As the correlations were not significant, the direction of the relationship is not clear, indicating that variability alone is not sufficient in determining breakup. Further research is needed to specifically investigate the nature of the relationship between emotion variance and breakup.

Table 1. Point-biserial correlations examining whether the emotional variance of someone's time series correlates with whether they break up.

Variance Break-up p-value

Male, positive interaction 0.10 .310

Female, positive interaction -0.06 .567

Male, conflict interaction 0.18 .086

Female, conflict interaction 0.16 .122

MINOR COMMENTS

MCC is a relatively uncommon measure for classification performance. For completeness the authors should add the standard metrics (accuracy, precision, recall, and AUC) to the visualizations and consider reporting AUC as the main metric throughout the paper.

ANSWER: We agree with the reviewer that for individuals familiar with machine learning, AUC is the better approach (amongst others which the reviewer mentioned) to measure classification performance. However, our experience has shown that psychologists are often unfamiliar with AUC, as it is not part of the standard statistical training. Although MMC is also not a standard tool in the psychologist's repertoire, we have found that psychologists have a better understanding of MMC, as they are familiar with the concept of correlation. Therefore, to ensure the results are understandable for psychologists, we will maintain MMC in the manuscript. Nevertheless, we have also calculated the standard metric of AUC and present it in the supplemental materials.

COMMENT 

The authors refer to their study as an experimental study on several occasions (e.g. page 21). This is imprecise as the study is clearly correlational and there was no randomly assigned manipulation. The authors should correct this.

ANSWER: In our opinion, it is an experiment because we have a manipulation - the interactions (positive interaction versus conflict interaction). Yes, there is no control group of other couples in this sense, but we can now compare the two interactions. This means that the control group for the positive interaction is the data from the conflict interaction and vice versa. This is the reason why we also compared then in this study. We believe that this is a very good control group and better than comparing it with other couples. Therefore, it is not actually an observation since we manipulated the interactions. We can change the terminology if the reviewer insists, but in our opinion, the term "observational data" is not correct in this case since we "forced" people into different interactions.

COMMENT 

Formatting / style

ANSWER: We have reformatted the manuscript and made it compatible with PLOS ONE guidelines.

COMMENT 

The following sentence is incomplete: “After classifying 10-second sequences as positive, negative, or neutral, they predicted partners’ next sequence the person’s prior sequence and found that unhappy couples experience greater reciprocity of negative affect (17,24) and high affect reciprocity 75 (12) than happy couples.”

ANSWER: We revised the sentence: 

” After classifying 10-second sequences as positive, negative, or neutral, the researchers predicted their partners' next sequence based on their prior sequence. They found that unhappy couples exhibit greater reciprocity of negative affect [17], [24] and higher affect reciprocity compared to happy couples [12].”

COMMENT 

The title on page 1 is different from the title on the cover page

ANSWER: Title: Emotion Dynamic Patterns Between Intimate Relationship Partners Predict Their Separation Two Years Later: A Machine Learning Approach

---

## [Editor Report · Decision Letter 1]

19 Jun 2023

Emotion Dynamic Patterns Between Intimate Relationship Partners Predict Their Separation Two Years Later: A Machine Learning Approach

PONE-D-22-33994R1

Dear Dr. Hilpert,

We’re pleased to inform you that your manuscript has been judged scientifically suitable for publication and will be formally accepted for publication once it meets all outstanding technical requirements. Congratulations on a fine paper!

Kind regards,

Joydeep Bhattacharya

Academic Editor

PLOS ONE
---

## [Editor Report · Acceptance letter]

23 Jun 2023

PONE-D-22-33994R1 

Emotion dynamic patterns between intimate relationship partners predict their separation two years later: A machine learning approach 

Dear Dr. Hilpert:

I'm pleased to inform you that your manuscript has been deemed suitable for publication in PLOS ONE. Congratulations! Your manuscript is now with our production department. 

Kind regards, 

on behalf of

Dr. Joydeep Bhattacharya 

Academic Editor

PLOS ONE